**An Intertropical Convergence Zone shift controlled the terrestrial material supply on the Ninetyeast Ridge**
*Xudong Xu[1,3,4], Jianguo Liu[1,2,4,*], Yun Huang[1,*], Lanlan Zhang[1,4], Liang Yi[5], Shengfa Liu[2,6], Yiping Yang[1,4], Li Cao[7], Long*
*Tan[1,3,4]*
[1]Key Laboratory of Ocean and Marginal Sea Geology, South China Sea Institute of Oceanology, Chinese Academy of
Sciences, Guangzhou 510301, China
[2]Laboratory for Marine Geology, Pilot National Laboratory for Marine Science and Technology, Qingdao 266061, China
[3]University of Chinese Academy of Science, Beijing 100049, China
[4]Southern Marine Science and Engineering Guangdong Laboratory (Guangzhou), Guangzhou 511458, China
[5]State Key Laboratory of Marine Geology, Tongji University, Shanghai 200092, China
[6]Key Laboratory of Marine Geology and Metallogeny, First Institute of Oceanography, Ministry of Natural Resources,
Qingdao 266061, China
[7]Shandong Provincial Research Institute of Coal Geology Planning and Exploration, Jinan 250104, China
Corresponding authors: Jianguo Liu (jgliu@scsio.ac.cn) and Yun Huang (huangyun@scsio.ac.cn)
**Abstract**
Among various climate drivers, direct evidence for the Intertropical Convergence Zone (ITCZ) control of sediment supply
on the millennial scale is lacking, and the changes in ITCZ migration demonstrated in paleoclimate records need to be
better investigated. Here, we use clay minerals and Sr-Nd isotopes obtained from a gravity core on the Ninetyeast Ridge
to track the corresponding source variations and analyze the relationship between terrestrial material supply and climatic
changes. On the glacial-interglacial scale, chemical weathering weakened during the North Atlantic cold climate periods
and falling sea level hindered the transport of smectite into the study area due to the exposure of Andaman and Nicobar
Islands. However, the influence of the South Asian monsoon on the sediment supply was not obvious on the millennial
scale. We suggest that the north-south migration of the ITCZ controlled the rainfall in Myanmar and further directly

determined the supply of clay minerals on the millennium scale because the transport of smectite was highly connected

with the ITCZ location; thus, the regional shift of the ITCZ induced an abnormal increase in the smectite percentage during

the late Last Glacial Maximum (LGM) in our records. The smectite percentage in the studied core is similar to distinct

ITCZ records but different in some periods, revealing that regional changes in the ITCZ were significantly obvious and

that the ITCZ is not a simple north-south displacement and closer connections occurred between the Northern-Southern

Hemispheres in the eastern Indian Ocean during the late LGM.

**1. Introduction**

Deposited sediments are essential recorders of the paleoclimate and paleoceanographic conditions since the climate is tied

to the whole sedimentation process from weathering and transport to the deposition of sediments on land. The terrestrial

materials of "source-sink" systems are supplied to marine environments under the combined effects of multiple climate-

related driving forces and ocean processes (Li et al., 2018; Yu et al., 2019), and understanding these effects is crucial for

reconstructing the coevolutionary relationship of the palaeoenvironment with the palaeoceanographic conditions and

palaeoclimate. Various factors may control the formation and transport of terrestrial materials at low latitudes, such as the

northeastern Indian Ocean. Recently, the South Asian monsoon has been revealed to be the main driving force of terrestrial

material supply in Bangladesh and of hydrological changes in the Bay of Bengal (BoB, Dutt et al. al., 2015; Gebregiorgis

et al., 2016; Joussain et al., 2017; Li et al., 2018; Liu et al., 2021). Moreover, the Intertropical Convergence Zone (ITCZ)

is an important climate-driving force in low-latitude regions (Deplazes et al., 2013; Ayliffe et al., 2013), which has a pivotal

role in heat transportation on Earth (Schneider et al., 2014), and the north-south shift of the ITCZ is thought to connect the

climates in the Northern and Southern Hemispheres (Huang et al., 2019; Zhuravleva et al., 2021). Because monsoon

dynamics are shaped by large-scale meridional temperature gradients and an ITCZ shift in tropical monsoon areas (Mohtadi

et al., 2016), there are hopeful opportunities to analyze sediment responses to ITCZ or monsoon variations. The

paleoclimate breakthroughs mentioned above enable us to analyze the response of sedimentary records to the ITCZ shift

in the BoB more accurately. However, evidence for direct control of terrestrial sediment supply by the ITCZ remains
lacking, which is an obstacle to understanding the response of the depositional environment to the ITCZ shift.

As the main deposition area for vast amounts of weathered Himalayan materials, the BoB accumulates numerous

Himalayan terrestrial materials that are loaded by the Ganges-Brahmaputra (G-B) River (Goodbred and Kuehl, 2000) and
forms the largest subaqueous fan, the Bengal Fan (3000 km long from north to south, 1400 km wide from east to west,
with an area of $3.9 \times 10^5$ km$^2$; Curray et al., 2003). The eastern and western sides of the BoB correspond to the Andaman
Sea and the Indian Peninsula, respectively, and the BoB is a natural site that is useful for studying the interactions between
weathering and climatic factors since both sides of the bay are affected by the South Asian monsoon (Ali et al., 2015).
Previous studies suggest that Himalayan material transported by the G-B River was the predominant source of material in
the northern BoB (Li et al., 2018; Ye et al., 2020), and the main sources in the west BoB are the Indian Peninsula and
Himalayan weathered material (Kessarkar et al., 2005; Tripathy et al., 2011; Tripathy et al., 2014). In the eastern BoB, the
sediment source areas include the Himalayan (transported by the G-B river), Indo-Burman Ranges and the Myanmar region
through which the Irrawaddy River flows (Colin et al., 1999; Joussain et al., 2016). The terrigenous detrital material in the
Andaman Sea is mainly Myanmar-origin sediments transported by the Irrawaddy River (Ali et al., 2015; Awasthi et al.,
2014; Colin et al., 2006). A series of terrigenous sediment issues, such as changes in the source area and proportion of
terrigenous matter in various regions of the BoB from the LGM to the Holocene, the distribution range of terrigenous
materials in the western and eastern BoB, and how the G-B River sediments are transported in the BoB, are unclear until
now. Little attention has been given to sediment provenance in the southern BoB or, particularly, to the correlation of these
sediment sources with climatic driving factors.

Recent studies have revealed that clay minerals can be used to effectively track changes in source areas in the source-

sink system of the BoB due to the great differences in clay mineral components among the source areas around the BoB
(Joussain et al., 2016; Li et al., 2017; Liu et al., 2019; Ye et al., 2020). Moreover, Sr-Nd isotopes have been widely reported
to track the variations in sediment provenance in the BoB (Ahmad et al., 2005; Colin et al., 1999; Colin et al., 2006).
In this study, we measured clay minerals and Sr-Nd isotopes in a deep-sea gravity core obtained from the southeastern
BoB (Figure 1) to reconstruct variations in the sources of sediments in the Ninetyeast Ridge and to further explore the
climate forces that affected the supply of terrestrial materials during the past 45 ka. Core 17I106 located above the abyssal
plain at ~900 m, exempting from the influence of large-scale turbidite activities and receiving only fine-grained pelagic
sediments that can reflect the changes in the provenance of the surrounding source area (Figure 1), which makes the
terrestrial sediments on the Ninetyeast Ridge suitable for exploring the relationship between the paleoclimate and
paleoenvironment in the BoB. Here, we aim to disentangle the ITCZ variability signal in marine sediments from multiple
driving forces and further understand the response of sedimentary records to ITCZ migrations.
**2. Materials and methods**
**2.1. Chronology**
Gravity core 17I106 (90.0040°E, 6.2105°N, water depth 2928 m) was collected by the *R/V Shiyan 1* vessel belonging to
the South China Sea Institute of Oceanology (SCSIO), Chinese Academy of Sciences (CAS), from the Ninetyeast Ridge,
northeast of the Indian Ocean (Figure 1). This core has a total length of 162 cm and consists of gray to green silty clays
subsampled at 1-cm intervals. The age model of core 17I106 was reconstructed based on 10 accelerator mass spectrometry
(AMS) [14]C dates and Bayesian interpolations between these dates (Figure 2 and Table 1). AMS [14]C dating was performed
on mixed planktonic foraminifera at Beta Analytic Inc. More than 20 mg of intact mixed planktonic foraminifera shells
were selected from the >150 μm fractions of each sample (10 g dried sample). All radiocarbon ages were converted and
reported as calendar years before present with the Calib8.2 software program with the Marine20 calibration dataset (Reimer
et al., 2020). A continuous depth-age model was performed using Bacon software by dividing a sedimentary sequence into
many thin segments and estimating a linear accumulation rate for each segment based on the calibrated [14]C dates and a
Bayesian approach (Blaauw and Christen, 2011).

**2.2. Clay mineralogy**

Clay minerals (<2 μm) were separated from the sediment samples by sedimentation according to Stokes' settling velocity
principle after organic materials and carbonates were removed with 15% hydrogen peroxide ($H_2O_2$) and 0.1 N hydrochloric
acid (HCl), respectively. We used the sedimentation method by placing the sample in glassware with an inner diameter of
7 cm and a height of 10 cm at an experimental temperature of 19 °C. The sedimentation time was calculated as 4 hours and
10 minutes according to Stokes' formula. The upper 5 cm of liquid was extracted, followed by centrifugation at 4800 rpm
for 10 minutes, and the smear was made into a natural slice. The natural slice was heated in an oven at 60 °C for 24 hours
to make ethylene glycol saturated slides for the subsequent test. The clay mineral slides were measured using routine X-
ray diffraction (XRD) equipment (Bruker Inc, D8 ADVANCE) in the Key Laboratory of Ocean and Marginal Sea Geology,
SCSIO, CAS. Clay mineral abundance was calculated by measuring the peak areas of smectite (15-17 Å), illite (10 Å) and
kaolinite/chlorite (7 Å). Relative proportions of kaolinite and chlorite were calculated from the ratio of 3.57 Å/3.54 Å peak
areas. The relative percentages of the four main clay minerals were estimated by calculating the integrated peak areas of
characteristic basal reflections using Topas5P software with the empirical factors by Biscaye (1965). The reproducibility
error of this method is ± 5-10%.

**2.3 Sr-Nd isotope analyses**

Twenty-two samples (<63 μm) from core 17I106 were selected for isotope analyses, and we used the experimental method
described by Dou et al. (2016). Carbonates were removed from 70 to 100 mg powdered bulk samples by leaching with
0.25 N HCl for 24 h at 50 °C. The residues were then completely digested in high-pressure Teflon bombs using a HCl +
$HNO_3$ + $HClO_4$ + HF solution. Rb and Sr were separated in 2.5 N HCl using Bio-Rad AG50W-X12, 200–400 mesh cation
exchange resin. Sm and Nd were separated in 0.15 N HCl using P507 cation exchange resin. The strontium (Sr) and
neodymium (Nd) isotopic compositions of the sediment samples were measured using a Thermo Scientific Multi-Collector
Inductively Coupled Plasma Mass Spectrometer (MC-ICPMS Nu plasma) at the Key Lab of Marine Sedimentology and
Environmental Geology, Ministry of Natural Resources, China. The organic materials and carbonate were removed from
the samples by $H_2O_2$ and HCl, respectively. For the convenience of direct comparison, the Nd isotopic ratio results are
expressed as $\varepsilon Nd$ (0)=[($^{143}Nd$/$^{144}Nd$)meas/0.512638-1]*10000, using the present CHUR value (Jacobsen and Wasserburg,
1980). Replicate analyses of NBS-987 during the study gave a mean $^{87}Sr$/$^{86}Sr$ of 0.710310 ± 0.000003 (2s), close to its
certified value of 0.710245. Similarly, replicate analyses of JNDi-1 gave a mean $^{143}Nd$/$^{144}Nd$ of 0.512112 ± 0.000004 (2s),
and its certified value was 0.511860.
**3. Results**
The age model is built based on 10 radiocarbon dates from core 17I106. The top age is 3.8 ka BP, and the bottom age is
44.9 ka BP; thus, this core covers a continuous sedimentary succession of the last ~45,000 years. The sedimentation rates
in the Holocene (average 3.1 cm/ka) were relatively lower than those during the last glacial period (average 4.6 cm/ka),
with the highest rate of 8.3 cm/ka during 12.5–13.6 ka BP (Figure 3a). In the study core, the illite percentage ranges from
31% to 63% with an average of 48%, while the smectite percentage ranges between 8% and 57% with an average of 30%
(Figure 3b-e). Moreover, the kaolinite percentage ranges from 2% to 16%, and the chlorite percentage ranges from 5% to
20% in the core sediments. In the study core, the $^{87}Sr$/$^{86}Sr$ ratios range from 0.7122015 to 0.7186141 with an average of
0.7161698, while $\varepsilon Nd$ values range from -13.02 to -10.29 with an average of -11.24 (Figure 3f-g). At this study core, the
$^{87}Sr$/$^{86}Sr$ ratio and $\varepsilon Nd$ values remain stable before the LGM but show fluctuations after the LGM, without obvious
increasing/decreasing tendencies. During ~14.5-12.5 ka, $^{87}Sr$/$^{86}Sr$ ratios significantly increased from 0.7139 to 0.7172,
while $\varepsilon Nd$ values decreased abruptly from -10.28 to -13.02.
**4. Discussion**

**4.1. Sediment provenance and transport patterns**

The lower sedimentation rates (3-5 cm/ka, Figure 3a) measured in core 17I106 were in accordance with the normal

sedimentation rates obtained from neighboring cores around the Ninetyeast Ridge (Ahmad et al., 2005; Raza et al., 2013).

In this region, turbidite activities were less developed (Joussain et al., 2016; Fournier et al., 2017), in accordance with its

far distance from the Active Channel (Figure 1). In the northern BoB, due to heavy river runoff and steep topography, the

G-B river system transports a large amount of the products of Himalayan physical denudation; these products mainly

consist of illite and chlorite formed under dry and cold climate conditions (Chamley, 1989; Khan et al., 2019). Because of

the hot and humid conditions in Myanmar and the Indian Peninsula, sediments in these regions are formed through the

chemical weathering of silicate minerals and thus have high smectite percentages. Moreover, the Irrawaddy River brought

weathered products characterized by high smectite percentages from Myanmar into the Andaman Sea, leading to high

smectite percentages in the terrestrial sediments deposited in this marine environment (Ali et al., 2015).

The relatively high illite percentages measured in core 17I106 indicate that the weathered Himalayan materials carried

by the G-B River system are the primary source of sediments in the study area (Figure 4a). Compared with the large

amounts of materials loaded by the G-B River system, the weathered areas and runoff volumes of the Indo-Burman Ranges

are relatively small, and consequently, their sediment contributions are limited in the study area, although their sediments

are also characterized by relatively high illite percentages (Joussain et al., 2016). Evidence of surface sediments in the BoB

further reveals that the smectite percentages of sediments in the central region are significantly lower than those in the

eastern and western regions (Li et al., 2017; Liu et al., 2019), indicating that sediments of Indian Peninsula origin are

difficult to transport into the eastern BoB through the central BoB. Because the limited weathering area of Andaman and

Nicobar Islands cannot provide a large amount of smectite according to provenance studies (Ali et al., 2015), the Myanmar

materials characterized by high smectite percentages have the advantage of shorter transport distances compared to those

sourced from the Indian Peninsula as the main source area of smectite around the BoB. Therefore, the most important

source of smectite in the study area is the Myanmar region. In marine environments, kaolinite is preferentially deposited
in estuary areas due to mineral segregation (Gibbs, 1977) and thus may not be transported over long distances, so the
kaolinite in the study area was most likely sourced from neighboring Sumatra (Figure 4a, Liu et al., 2012). The Sr-Nd
isotopes measured in the studied core are close to those measured in the Irrawaddy/Indo-Burman Ranges/Sumatra source
regions (Figure 4b), indicating that terrestrial materials with diameters <63 μm mainly come from the Irrawaddy River,
Indo-Burman Ranges and Sumatra source areas, which was confirmed by a Sr-Nd isotope study in the southwestern part
of the study area (Ahmad et al., 2005) and consistent with sediment provenance studies in the Ninetyeast Ridge on different
timescales (Ali et al., 2021; Seo et a., 2022). This difference in clay minerals and isotopes may be consistent with the view
that clay minerals may be transported over long distances, while coarser terrestrial sediments can only be transported to
more proximate locations.
In the northeastern BoB, the southwest monsoon turns southward into the Andaman Sea, resulting in the transport of
sediments from the Indo-Burman Range and Irrawaddy River to the central Andaman Sea (Colin et al., 2006). The location
of core 17I106, drilled on the Ninetyeast Ridge, was above the abyssal plain, and the terrestrial materials deposited to the
west of this location are difficult to resuspend and deposit on the ridge under the force of bottom currents or turbidity
currents. In fact, the G-B River-loaded materials are mainly carried eastward by surface ocean currents in summer to the
Andaman Sea, where the seasonal surface currents load materials from the Himalayan and Indo-Burman Ranges into the
Andaman Sea through the northern strait (NS) (Liu et al., 2020a; Rayaroth et al., 2016). These G-B River sediments can
also be transported southward along the west side of the Andaman and Nicobar Islands (Figure 5), and a westward ocean
surface current in the middle strait (MS) loads sediments of the Irrawaddy River southwest into the study area (Chatterjee
et al., 2017).
**4.2. Factors affecting sediment provision**
In general, illite is the major mineral produced during the strong physical erosion of metamorphic rocks and granite rocks
and during the reprocessing of sedimentary rocks (Chamley, 1989; Winkler et al., 2002), while smectite is the secondary
mineral produced during the chemical weathering of parent aluminosilicate and iron-magnesium silicate under warm and
humid climate conditions (Chamley, 1989; Erosion, 1995). The climatic forces from the North Atlantic are thought to
extensively impact the tropical Eastern Indian Ocean (EIO) and surrounding areas of the BoB (Sun et al., 2011; DiNezio
and Tierney, 2013; Dutt et al., 2015; Gautam et al., 2020; Mohtadi et al., 2014; Liu et al., 2021), whose climate signals can
be transmitted via the tropical Atlantic bipolar SST anomaly and associated southward shift of the ITCZ (Marzin et al.,
2013), westerlies teleconnection and sea ice (Sun et al., 2011) or the reorganization of the Hadley circulation (Mohtadi et
al., 2014). During the North Atlantic cold-climate periods (Heinrich events and YD period, Figure 3h), when rainfall and
temperatures decreased in the South Asian monsoon region (An et al., 2011; DiNezio and Tieryney, 2013; Gautam et al.,
2020), physical weathering was enhanced in the Himalayas (Joussain et al., 2016), which made illite percentages at core
17I106 relatively high during these cold-climate periods, but chemical weathering weakened in Myanmar, and the smectite
percentage thus decreased in the source area before these cold periods and continued to increase after these periods. The
increasing (decreasing) trend of illite (smectite) percentages before cold-climate periods and the decreasing (increasing)
trend of illite (smectite) percentages after cold-climate periods in our records suggest that the weathering degree in the
source area influenced the supply of clay minerals during these cold-climate periods.

Sea level fluctuation is also critical in controlling the supplementation of terrestrial materials, especially clay minerals

(Li et al., 2018; Liu et al., 2019), by changing the transport paths and/or distances as well as the further input of sediments
into the study area. The changing trends of the sea level in seas adjacent to the BoB (Figure 3i, Waelbroecka et al., 2002;
Grant et al., 2014; Hanebuth et al., 2000; Thompson and Goldstein, 2006) are well correlated with the smectite percentages
measured in core 17I106, especially during 35-21 ka, when the smectite percentages declined continuously. Since the
Andaman and Nicobar Islands connecting the Andaman Sea and the BoB have continuously expanded as the sea level has
continuously declined, the strait width has been consistently reduced, thereby preventing the entrance of terrestrial
materials into the Andaman Sea and the further continuous decline in smectite percentages in the study area. Here, we
suggest that the variations in the measured illite percentages were mainly caused by changes in smectite deposition because
the sedimentary records obtained from the northern BoB do not support the controlling effect of sea level on illite
percentages over the past 50 ka (Joussain et al., 2016; Li et al., 2018; Liu et al. al., 2019). The relative exposure of 200 km
from the current Irrawaddy River delta may affect the deposition process on the continental shelf or further deposition of
the sediments delivered to the deep ocean, but core 17I106 is formed by the long-distance transport of large amounts of
fine-grained terrestrial material, indicating that these sediments can be transported over long distances, and the ~200 km
change in the shelf distance is not a dominant factor of sediment transport in the study area. Moreover, the decreasing
smectite percentages from the Myanmar area as sea level decreases suggests that shelf denudation is also not the main
factor affecting our smectite record, which is in accordance with previous studies in the Andaman Sea that have not
specifically emphasized the alteration of terrestrial source material supply by exposed shelves (Ali et al., 2015; Awasthi et
al., 2014).
The South Asian summer monsoon is normally thought to be an important factor affecting weathering conditions
around the BoB (Dutt et al., 2015; Gebregiorgis et al., 2016; Joussain et al., 2017; Li et al., 2018; Rashid et al., 2011; Zhang
et al., 2020; Zorzi et al., 2015). Stalagmites in Mawmluh Cave record variations in river runoff in the surrounding area;
these variations are determined by the impacts of surface sea temperature (SST) and water vapor transport paths (Dutt et
al., 2015). In fact, the Mawmluh Cave records of the South Asian monsoon strength are driven by temperature gradients
that drive changes in winds and moisture transport into the BoB (Dutt et al., 2015), not just responding to the rainfall
amount. The smectite percentage changes measured in core 17I106 were slightly correlated after Heinrich event 1 (H1) but
were irrelevant before H1 (Figure 6b). This indicated that the combination of temperature and moisture failed to play a
crucial role in smectite transport to core 17I106, although weathering features in the source area may be shaped by the
South Asian monsoon. Moreover, the view could be confirmed by the smectite record obtained from the studied core not
being well correlated with records previously obtained in the Andaman Sea (Figure 6c, 6d, Gebregiorgis et al., 2016) or
with a sporopollen record obtained in Southwest China (Figure 6e, 6f, Zhang et al., 2020), especially before the LGM. The
consistency of salinity and SST in core SK 168 (Figure 6c, 6d) and moisture and temperature index (Figure 6e, 6f) in
Southwest China reveal that the hydroclimate in the South Asian monsoon region might have been influenced by SST in
the Indian Ocean. All these inconsistencies between the smectite percentage in core 17I106 and monsoon records indicate
that smectite supplementation may be mainly controlled by rainfall rather than by chemical weathering due to
thermodynamic differences between sea and land environments (Liu et al., 2020b).

During the late LGM, the smectite percentage increased abnormally in core 17I106, and this increase cannot be

explained by dry and cold weathering conditions, a lower sea level or a weakened summer monsoon at that time. In contrast,
this abnormal change may have been attributed to an increase in the smectite input in sediments from the Burman source
area or to a decrease in the amounts of sediment input from the Himalayas. Under the influence of the winter monsoon
during the LGM, the denuded sediments on the Irrawaddy estuary shelf may have been transported southward through the
west side of Andaman Island (Prajith et al., 2018), as was confirmed in previous work showing that the winter monsoon
led to an increase in terrestrial materials from the Irrawaddy River to the Ninetyeast Ridge during the Heinrich event
(Ahmad et al., 2005). However, the winter monsoon was strong in the western part of the study area from 21 to 15 ka
(Figure 6g), and the sea level remained relatively low during that period (Gautam et al., 2020). The smectite percentages
in the studied core increased significantly from 21 to 19 ka and dropped rapidly after 19 ka. This inconsistency contradicts
the conclusion that the increased smectite percentage in the source area was caused by a strong winter monsoon. Moreover,
the changes in the sediment compositions measured in the Himalayan source area were probably related to variations in
regional glaciers. During the LGM period, the increased glacial cover may have reduced surface runoff and further the
transport of physical weathering products, while the increased amount of ice meltwater may have transported more illites
following glacial melt. However, the reduced glacial area in the Himalayas during 18-15 ka did not occur simultaneously
with the increased illite percentage (Yan et al., 2020; Weldeab et al., 2019, Figure 6h). Therefore, the abnormal changes
measured in the smectite percentage during the late LGM period were caused by other climate-driven mechanisms, and the
millennium-scale smectite percentage fluctuations that occurred before the LGM require a more reasonable explanation.

**4.3. The ITCZ shift in the EIO**

Changes in rainfall and the corresponding runoff are generally utilized to explain short-term variations in clay minerals. In
the EIO, rainfall is controlled by monsoon activities (An et al., 2011; Beck et al., 2018; Gebregiorgis et al., 2016) and/or
ITCZ migrations (Deplazes et al., 2013; Stoll et al., 2007; Tan et al., 2019). Glacial-interglacial monsoon precipitation
changes at the regional scale are shaped by dynamics (changes in the wind fields) and temperature (McGee, 2020). The
wind fields may be driven by the relative dominance of the northern low-pressure and southern high-pressure systems (An
et al., 2011) and cross-equatorial moisture transport (Clemens et al., 2021), while the SST in the eastern Indian Ocean
(Zhang et al., 2020) or western Indian Ocean (Wang et al., 2022), surface and subsurface temperature changes (Tierney et
al., 2015), and temperature gradients (Weldeab et al., 2022) also play an important role in South Asian rainfall. At the same
time, as a climate-driving force in low-latitude regions, ITCZ migrations may be the main factor responsible for regional
hydrological changes (Deplazes et al., 2013; Weber et al., 2018) since the shift in the ITCZ was considered to control
rainfall distribution and intensity in central India over geological time scales (Zorzi et al., 2015) and to cause summer
temperature and moisture fluctuations in southwestern China during the last deglaciation (Zhang et al., 2019).
During the glacial-interglacial period, the ITCZ migrated north-south and balanced thermal differences by transferring
atmospheric heat; this process represents an indispensable climate-regulating power on Earth (Broccoli et al., 2006; McGee
et al., 2018; Schneider et al., 2014). In the Cariaco Basin and Arabian Seas (Figure 7a-b), tropical rainfall is highly
correlated with the North Atlantic climate, and sea ice variations in the North Atlantic affect the north-south shift of the
ITCZ in low-latitude regions through atmospheric circulation and ocean processes (Deplazes et al., 2013). The smectite
particles measured in core 17I106 mainly came from the Myanmar source area; in this area, rainfall is greatly affected by
the seasonal shift of the ITCZ. Before the LGM, the smectite percentages in the study core were well matched with the
ITCZ record in the Arabian Sea, where the supplementation of smectite percentages reached the peak when the ITCZ
shifted significantly northward (Figure 2b; Deplazes et al., 2013). During cold climate events, when the ITCZ moved
significantly southward, rainfall decreased, and the smectite percentages decreased correspondingly in the source area.
Therefore, we suggest that these changes in the smectite percentages in the studied core are correlated with ITCZ migration
and that rainfall is an important factor determining the smectite percentage from the source area of Myanmar on the
millennial scale. If precipitation induced by wind and temperature of the South Asian monsoon have an intense impact on
the source area, the source area monsoon indicators, for example, foraminifera, sporopollen, stalagmite (Figure 6) and other
indicators, would correspondingly change, but our record failed to catch these variations in monsoon indicators in the BoB.
We suggest that every factor affecting precipitation induced by wind and temperature of the South Asian monsoon, as
mentioned above, may have made it difficult to cause millennial-scale fluctuations similar to the ITCZ shift during the
MIS3 period. The South Asian monsoon is indeed the result of combined factors that may contribute to the heterogeneity
of monsoon rainfall in the BoB, which were also influenced by the north-south shift of the ITCZ. In core 17I106, the
corresponding variations in the relatively high smectite percentages and the northward shift of the ITCZ indicate that the
northward movement of the ITCZ is the most important factor influencing the incremental changes in river sediment load
corresponding to the increased smectite percentages in the Myanmar region. Here we emphasize that the northward and
southward ITCZ shifts bring about rainfall increases and decreases relative to other rainfall forces. The changes in clay
minerals reflect changes in clay mineral supply in the source area, and it is that these relative increases and decreases in
rainfall lead to changes, which is a response to environmental changes. The sporopollen evidence suggested a cold and wet
period during MIS3 in Yunnan, China (Zhang et al., 2020), which may have been caused by the frequent northward
movement of the ITCZ during this period.
Although the changes in smectite percentages in the study area are associated with ITCZ shifts before the LGM, the
ITCZ shift in the Indo-Pacific warm pool (IPWP) was more "regional" than those in the Arabian Sea and the Cariaco Basin
(Deplazes et al., 2013). During the late LGM, when the ITCZ did not move extensively in the Arabian Sea, the ITCZ
gradually shifted northward in the IPWP from 21-18 ka (Figure 7d, Ayliffe et al., 2013). However, the smectite percentage
increased significantly in the study area, and we have excluded the possibility that the winter monsoon or meltwater
influenced these changes. Further comparisons with IPWP records reveal that the ITCZ changes agree well with the
smectite percentage variations during the late LGM, indicating that the northern migration of the ITCZ induced high
smectite percentages in core 17I106 (Figure 7c, d). These results suggest that the clay minerals of core 17I106 are
inextricably linked to ITCZ shifts on the millennial scale. In summary, our smectite record shows that before the LGM, the
ITCZ was in a relatively southerly position in the Myanmar area, while during the late LGM, the northward movement of
the ITCZ in the BoB led to increased rainfall in the Myanmar source area and an increased supply of smectite. At the same
time, the ITCZ was not significantly shifted in the Arabian Sea region either pre-LGM or post-LGM, which is what the
Arabian Sea record shows (Deplaze et al., 2013).

The smectite percentage in the studied core is different from the ITCZ records in some periods, such as the late LGM,

revealing that regional changes in the ITCZ were significantly obvious and that the ITCZ is not a simple N-S displacement.
This consistency may indicate that the regional extension of the north-south thermodynamic gradient in the EIO exceeded
that in the Arabian Sea and that the north-south shift of the ITCZ caused the climate systems of the Northern and Southern
Hemispheres to be more closely connected in the EIO during the late LGM (Huang et al., 2019; Zhuravleva et al., 2021).
A recent study considered less northward migration of the summer ITCZ position in the western BoB than in the eastern
BoB during Heinrich Stadials HS1 and HS5 (Ota et al., 2022), which indicated that regional ITCZ variations in the BoB
may be very common. These factors may be correlated with observed variations in regional air-sea interactions, such as
the exposure of the Sunda Shelf (DiNezio and Tierney, 2013), the effect of the thermocline in the EIO (Mohtadi et al., 2017)
and even potential El Nino-like mode (Thirumalai et al., 2019) and IOD (Abram et al., 2020) changes, which may make
the ITCZ shift more dramatic or keep the ITCZ position in the Northern Hemisphere longer. Thus, the regional variations
in the ITCZ should be fully considered when studying climate change, especially in low-latitude regions that are sensitive
to climatic and environmental changes, such as the EIO (Niedermeyer et al., 2014).

**5. Conclusion**

We reconstructed the variations in sediment sources on the Ninetyeast Ridge over the past 45 ka. The main source areas
comprise the Himalayas transported by the G-B River and Irrawaddy River; sediments were stably supplied from these
regions throughout the studied core. When North Atlantic cold events (Heinrich and YD) occurred, chemical weathering
weakened and physical weathering increased; correspondingly, the smectite percentage decreased and the illite percentage
increased. From 35-21 ka, the falling sea level led to an increase in the exposed area of the Andaman and Nicobar Islands
and further hindered the entrance of smectite from the Andaman Sea into the study area. At the same time, the influence of
the South Asian monsoon on the sediment supply was not obvious. The time-phase mismatches observed among records
excluded the influence of Burman shelf sediment erosion forced by the winter monsoon or of variations in G-B river
sediments induced by ice meltwater on the abnormal increases observed in the smectite percentages during the late LGM.
The smectite record of core 17I106 is consistent with the ITCZ changes recorded on the millennial scale, indicating that
the ITCZ controls the rainfall in the Burman source area and, further, the clay mineral variations in the study area. The
inferred ITCZ shift recorded in the studied core coincided with the global ITCZ change that occurred before the LGM, but
during the late LGM, the core record was consistent with the change in the regional ITCZ recorded by the IPWP. This
revealed that regional changes in the ITCZ were very significant, and the ITCZ is not a simple N-S displacement at the
same time. Thus, the regional variations in the ITCZ should be fully considered when studying climate change, especially
in low-latitude regions that are sensitive to climate and environmental changes.

**Author contributions.**

J.L. and Y.H. conceived and designed the experiment. X.X. wrote the manuscript with contributions from all authors. L.Z.
and L.Y. provided the ages of planktonic foraminifera, and S.L., Y.Y., L.C., and L.T. helped to analyze the measured data
and discuss the related relevant topics in this manuscript.
**Competing interests.**
The authors declare that they have no conflicts of interest.
**Acknowledgements.**
We thank Hui Zhang for the help of Sr-Nd isotope measurements. We also thank the editor Pierre Francus, Prof. Michael
E. Weber and another anonymous reviewer for their constructive comments. Core sediment samples were collected onboard
of *R/V "Shiyan 1"* implementing the open research cruise NORC 2012-08 supported by the NSFC Shiptime Sharing Project.
**Financial support.**
This work has been jointly funded by the National Natural Science Foundation of China (42176075, 42130412 and
41576044), Key Special Project for Introduced Talents Team of Southern Marine Science and Engineering Guangdong
Laboratory (Guangzhou) (GML2019ZD0206), and the Open Fund of the Key Laboratory of Submarine Geosciences,
Ministry of Natural Resources (KLSG2102).
**Data Availability Statement.**
All dataset is available on Science Data Bank
(https://www.scidb.cn/detail?dataSetId=55c7dcf1f8344c658099dfe030264b2f).

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

**Figure Captions**

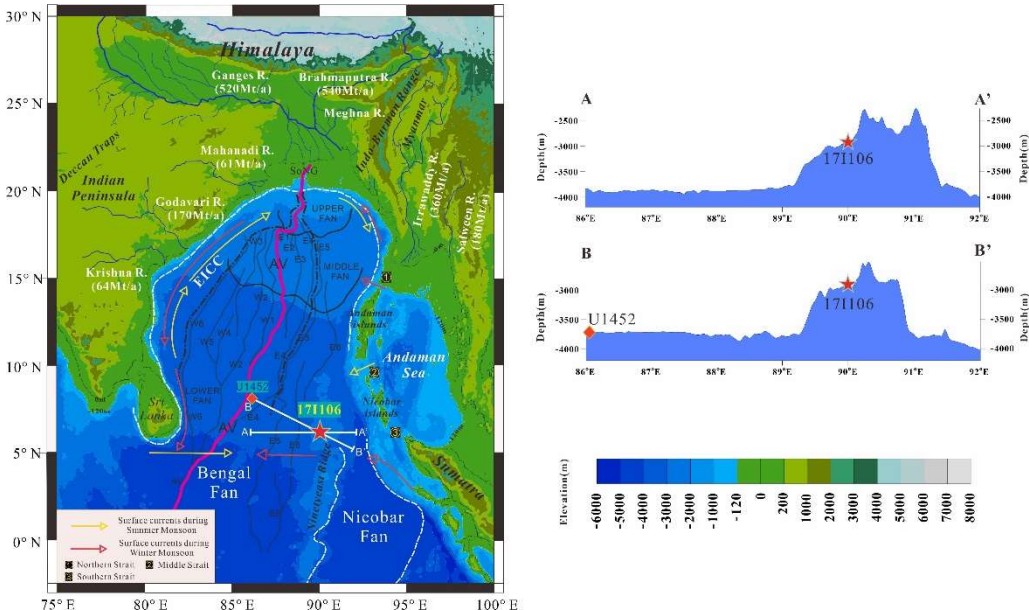


**Figure 1.** Geographical setting of the BoB. The locations of cores 17I106 (red asterisks) and U1452 (orange diamond) are
shown. On the left, the white dashed lines outline the scale of the Bengal Fan and the Nicobar Fan. The pink solid line is
the "active" channel, and solid gray lines and black letters represent the turbidity channel and the reference names of the
principal channels. The dotted-dashed line is the outline of the most recently active subfan (Curray et al., 2003). The solid
white lines denote the two profile positions, which are shown on the right with the elevation legend.

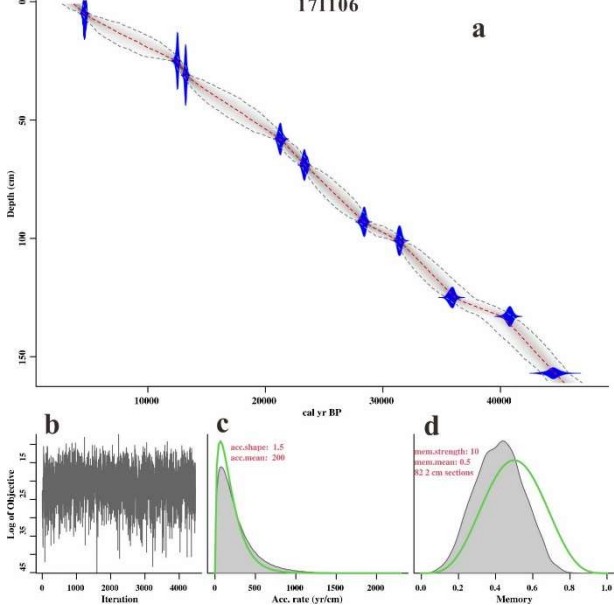


**Figure 2.** Age-depth model of core 17I106 in the northeastern Indian Ocean. **a,** Calibrated [14]C dates (blue, with 2σ errors)
and the resulting age-depth model (the darker gray shading indicates more likely calendar ages; the gray stippled lines
show 95% confidence intervals; and the red curve shows the single 'best' model based on the weighted mean age for each
depth). **b,** Number of Markov chain Monte Carlo (MCMC) iterations used to generate the grayscale graphs. **c,** Prior (green)
and posterior (gray) distributions of the sediment accumulation rates (the mean sediment accumulation rate was ~2
years/cm). **d,** Prior (green) and posterior (gray) memory distributions (dependence of the sediment accumulation rate
between neighboring depths).

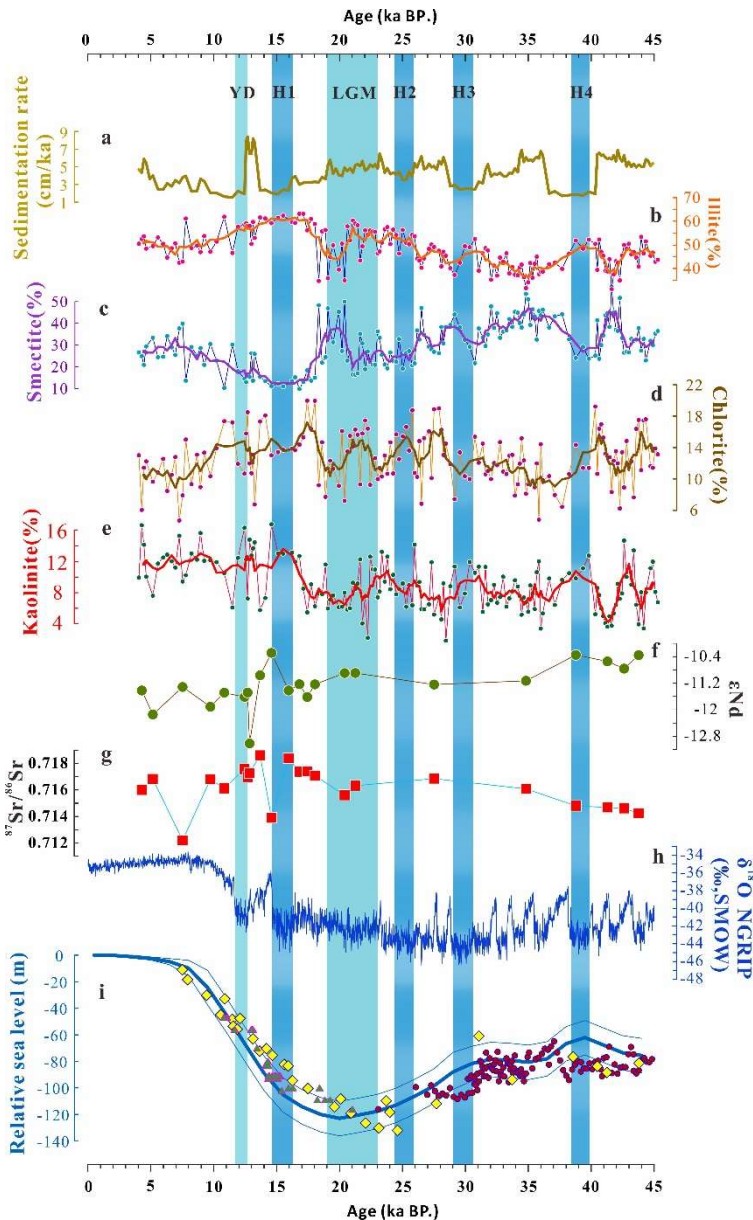


**Figure 3.** Comparison of clay mineral and Sr-Nd isotopes data in the northeastern Indian Ocean with paleoclimate records.

**a**, Sedimentation rate in core 17I106; **b, c, d, e**, illite, smectite, chlorite and kaolinite percentages in core 17I106 (thick line

represents a 3-point running average); **f, g** $^{87}Sr/^{86}Sr$ and εNd values of core 17I106 in the northeastern Indian Ocean; **h,**
δ$^{18}$O data of Greenland ice core NGRIP (Svensson et al., 2008); **i,** Global sea level as proxy for ice volume, reconstructed
from benthic δ$^{18}$O (thick cyan line, thin cyan line represents the 95% confidence interval, Thompson and Goldstein, 2006),
globally distributed corals (yellow dots, Waelbroecka et al., 2002) and sea level data (Triangles and red dots) collected by
Grant et al.(2014) and Hanebuth et al. (2000). Blue and cyan bars represent cold climate periods of Heinrich events (H1-
H4) together with Younger Dryas (YD) and the last glacial maximum (LGM), respectively.

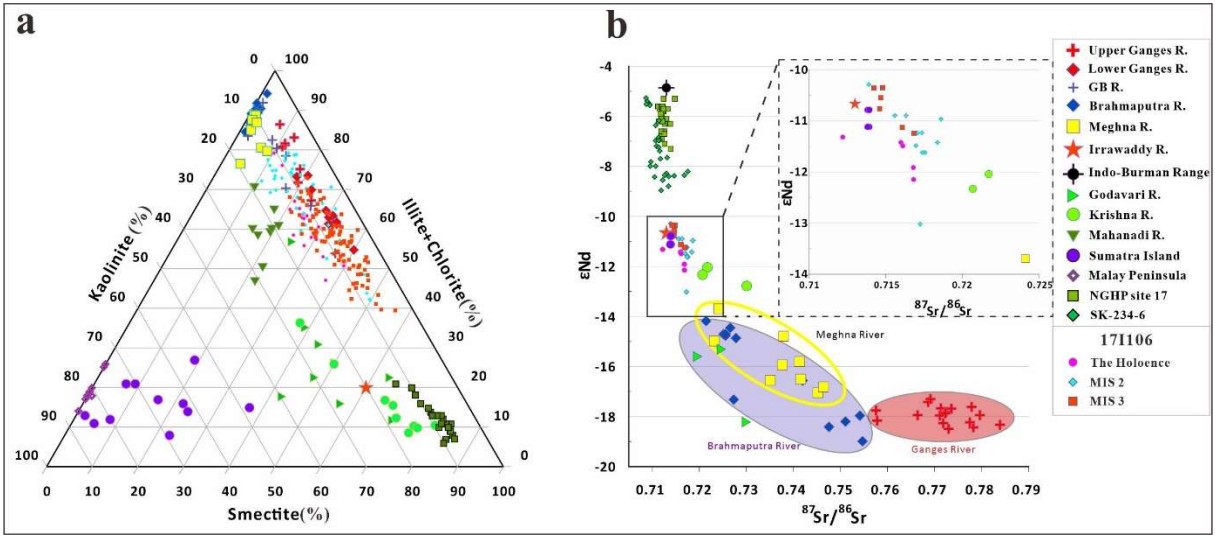

**Figure 4.** Sediment provenance of core 17I106 in the northeastern Indian Ocean. **a,** Sediment provenance discrimination
diagram in the northeastern Indian Ocean. For comparison, clay mineral data obtained from sediments collected in the
modern Ganges River, Brahmaputra River Lower, Ganges-Brahmaputra River Lower and Meghna River (Khan et al., 2019),
Mahanadi and Krishna Rivers of Indian Peninsula (Bejugam and Nayak, 2017), Irrawaddy River (Rodolfo, 1969), and
Sumatra and Malay Peninsula rivers (Liu et al., 2012) are also plotted. The referenced cores comprise NGHP Site 17 (Ali
et al., 2015), representing the Irrawaddy River as the main clay mineral source in the Andaman Sea. **b,** Variations in εNd
(0) vs. $^{87}Sr/^{86}Sr$ measured in core 17I106 compared with those measured in river sediments and bulk rock samples collected
around the BoB. In this diagram, we display data collected from Indian river samples (from the Godavari and Krishna
Rivers) (Ahmad et al., 2009) from different parts of the modern G-B River system (Lupker et al., 2013). Measurements
taken from sediments obtained from the Irrawaddy River (Colin et al., 1999), formations from the Indo-Burman ranges
(Licht et al., 2013) and volcanic products of Sumatra Island (Turner et al., 2001) are also plotted. The referenced cores
include NGHP Sites 17 and SK-234-60, both of which indicate that the Irrawaddy River is the main Sr-Nd isotope source
for the Andaman Sea.

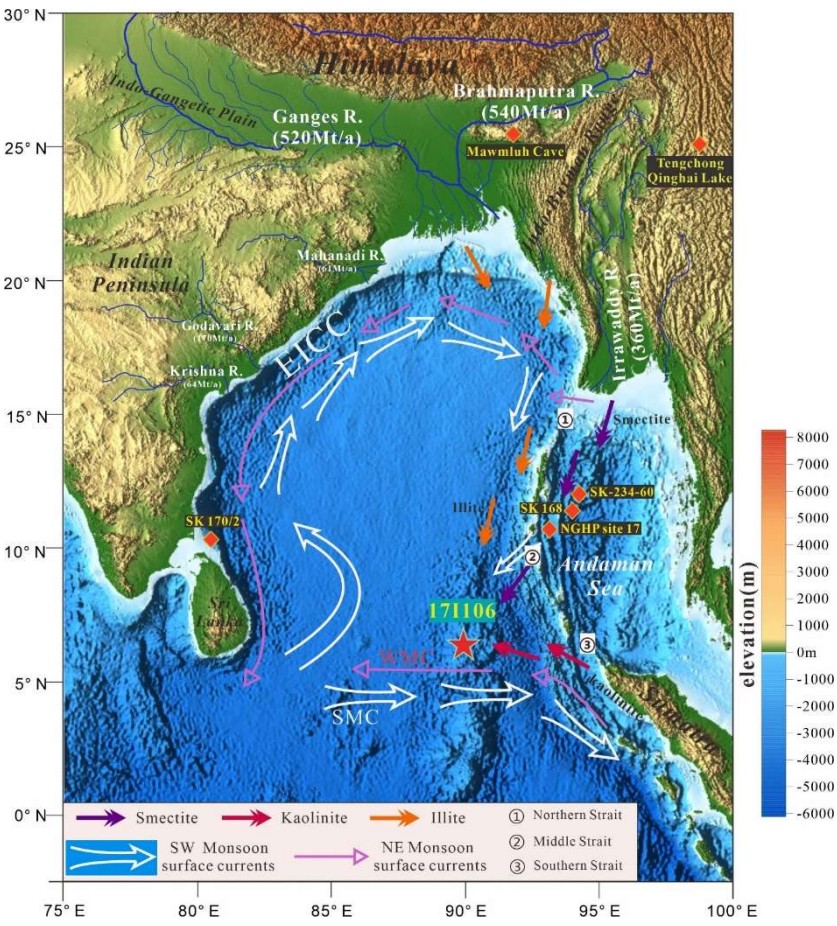

**Figure 5.** Map showing dispersal patterns of the BoB clay minerals for core 17I106. The locations of core 17I106 (red
asterisks) and of the reference core and sites are shown: SK 170/2 in the northern BoB, SK-168, SK-234-60, NGHP site
17 in the western Andaman Sea, and Mawmluh Cave in northeastern India and Tengchong Qinghai Lake in China are
represented by orange diamonds. The orange, purple and red arrows represent the main dispersal directions of illite,
smectite and kaolinite when the fluvial sediments were discharged into core 17I106. The white and red arrows denote the
SW and NE monsoon currents, respectively. In the western BoB, the East Indian Coastal Current (EICC) reverses annually
with the monsoon wind (Schott and McCreary, 2001). In the lower-latitude regions of the BoB, monsoon-driven currents
flow eastward in summer to form the summer monsoon current (SMC) and westward in winter to form the winter monsoon

 current (WMC) (Shankar et al., 2002). The elevation legend is shown to the right of this figure.

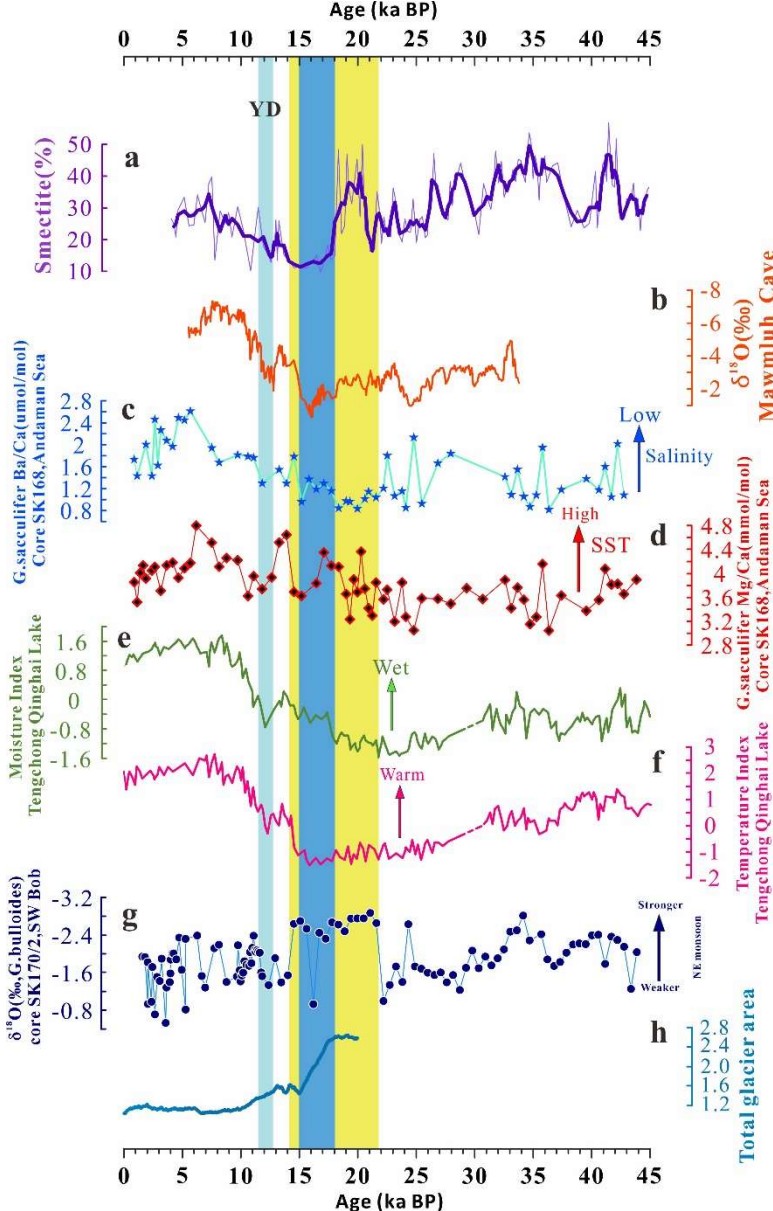


**Figure 6.** Comparison of smectite percentages in core 17I106 with paleoclimate records. **a**, Smectite percentages in core

17I106 (thick line represents a 3-point running average); **b,** Mawmluh Cave $\delta^{18}$O record for the interval 33,800 to 5500

years BP (Dutt et al., 2015). **c, d,** Ba/Ca and Mg/Ca of the mixed layer species *G. sacculifer* in core SK 168 from the

Andaman Sea, which represent the surface sea salinity and temperature, and the lower salinity and higher temperature

showed a strong SW monsoon (Gebregiorgis et al., 2016). **e, f,** Moisture index and temperature index from pollen records

from Tengchong Qinghai Lake, respectively (Peng et al., 2019; Zhang et al., 2020). **g,** $\delta^{18}$O variability record of planktic

foraminifera *Orbulina universa* obtained from core SK 170/2 recovered from the southwestern Bay of Bengal, which
represents the strength of the NE monsoon (Gautam et al., 2020). **h,** Ratio of the modeled total glacier area over the southern
parts of the Himalayan-Tibetan orogen to the present level (Yan et al., 2020). Yellow, blue and cyan bars represent the
strong NE monsoon period shown by line **g**, the main periods of glacier melting in the southern Himalayas shown by line
**h** and the cold climate periods of the Younger Dryas (YD).

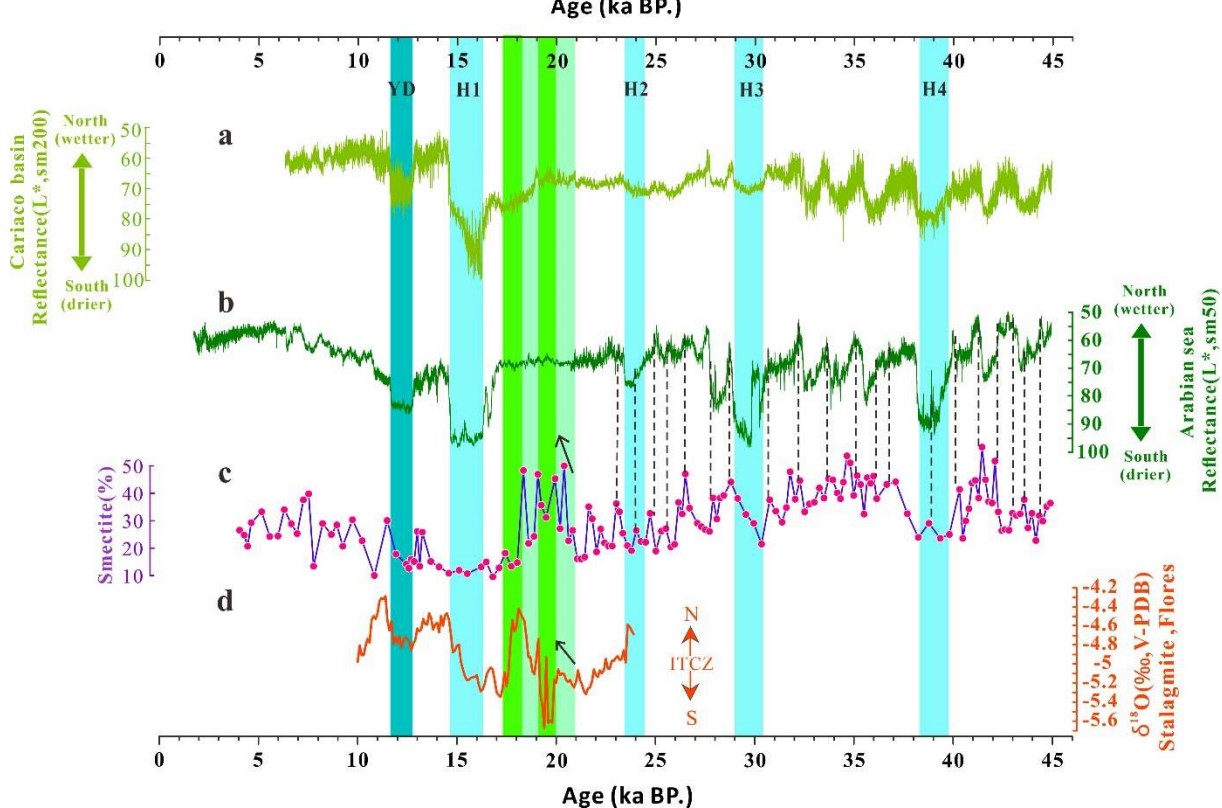

**Figure 7.** Comparison of smectite percentages with ITCZ north-south shift records. **a,** L∗ represents the ITCZ shift from

the Cariaco Basin (Deplazes et al., 2013); **b,** L∗ represents the ITCZ shift from the Arabian Sea (Deplazes et al., 2013); **c,**

Smectite percentages in core 17I106; **d,** Stalagmite $\delta^{18}O$ record from Flores (Ayliffe et al., 2013). The gold dotted line

denotes the connection between the northward movement of the ITCZ and the peak smectite percentage, and the series of

color bars from 21-18 ka represent the ITCZ-shift periods recorded in **d.** The green bars represent the consistent periods

shown in **c** and **d** in the late LGM, and the black arrows in **c** and **d** indicate great differences between the smectite

percentages and ITCZ record in the EIO.

**Table 1.** Carbon-14 and calibrated calendar ages of mixed planktonic foraminifera measured in core 17I106 in the

northeastern Indian Ocean.

| Number | Depth (cm) | Materials | Measured $^{14}$C age (yr BP, ±1σ) | Calendar median age (yr BP) |
|---|---|---|---|---|
| 1 | 5 | mixed planktonic foraminifera | 4160±30 | 4053 |
| 2 | 25 | mixed planktonic foraminifera | 10690±40 | 11880 |
| 34 | 31 | mixed planktonic foraminifera | 11460±40 | 12801 |
| 4 | 58 | mixed planktonic foraminifera | 17910±50 | 20710 |
| 5 | 69 | mixed planktonic foraminifera | 20050±60 | 23183 |
| 6 | 93 | mixed planktonic foraminifera | 24590±90 | 27883 |
| 7 | 101 | mixed planktonic foraminifera | 27820±120 | 31074 |
| 8 | 125 | mixed planktonic foraminifera | 31820±200 | 35455 |
| 9 | 133 | mixed planktonic foraminifera | 36370±280 | 40434 |
| 10 | 157 | mixed planktonic foraminifera | 42190±560 | 44167 |

656