# Peer review of "An Intertropical Convergence Zone shift controlled the terrestrial material supply on the Ninetyeast Ridge"

_Climate of the Past, 2021_

## Author Comment (AC2)

**comments (black font) and responses (blue font)**

Reviewer #2
General comments:

Xu et al. present a very interesting and useful sediment record from Ninetyeast Ridge with measurements of clay mineralogy and Sr-Nd isotopes. Although the time frame of the record is somewhat limited, it does provide some constraints on orbital variation of terrigenous sediment fluxes. The data are of good quality and the interpretations are reasonable. The paper is well within the scope of Climate of the Past and provides another helpful record of the South Asian monsoon, which drivers are important geological questions. Broadly, I think the paper would be improved by clarifying the uncertainties in the interpretation of the proxies, more information about methods, and interpretation of the driving factors (ITCZ versus other influences on precipitation intensity, weathering and sediment transport).

Specific Comments:

Please add more perspective on disentangling the transport limitations (trapping of sediments on the shelf) and weathering (ITCZ migration and rainfall intensity).

How might the transport conditions mask the ITCZ record (e.g. lower smectite transport is interpreted as lower rainfall). It's great that both factors are mentioned, but I think the text needs to be clarified as to how these factors may result in different interpretations of the same record. This would give more confidence in the rainfall/ITCZ related interpretations.

I think it is important to discuss the potential factors on rainfall and weathering other than north-south ITCZ migration. For instance, Gebregiorgis et al. 2016 QSR suggest that drivers off the South Asian monsoon may be more complex than migration of the ITCZ.   There has been recent work regarding exposure of the Sunda/Sahul Shelves on Indian Ocean hydroclimate (DiNezio and Tierney, 2013 Nat. Geosci; DiNezio et al., 2020 Sci. Adv.; Pico et al., 2020 Paleoc. Paleoclim.), cross-equatorial moisture transport and influence on the monsoon (Clemens et al., 2021, Sci Adv.), the effect of stratification on the monsoon (Tierney et al. 2015, Nat. Geosci), and even a potential El Nino-like mode in the Indian Ocean during the LGM (Thirumalai et al., 2019 Paleoclim. Paleoc.). These are all factors that can influence precipitation that are not necessarily related to just the position of the ITCZ. There is mention of some potential other factors on lines 235-242, but this is not clear and not discussed directly related to your record. It would be good to consider your smectite record in regard to the timing of these other processes; a clearer discussion of these factors would improve the relevance of what sediment flux records like this are actually recording the Indian Ocean.

Re: Thank you for your helpful comments, and we would be happy to add a discussion of other potential factors to the text. We believe that the exposure of the Sunda Shelf (DiNezio and Tierney, 2013; DiNezio et al., 2020; Pico et al., 2020), trans-equatorial water vapor transport (Clemens et al., 2021), surface and subsurface temperature changes (Tierney et al. 2015), Indian Ocean surface temperature gradient(Weldeab et al., 2022) and even ENSO (Thirumalai et al., 2019) and IOD (Abram et al., 2020) changes, if they have an impact on the source area, should lead to variations in the connected indicators, for example, foraminifera, sporopollen, stalagmites and other indicators. Although there are some disturbances from the limitations of various monsoon indicators, the South Asian monsoon is indeed the result of a combination of factors that may contribute to the

differences in monsoon rainfall in the Arabian Sea and the BoB, which is why the monsoon intensity variability shown by the monsoon indicators is not the same. The north-south shift of the ITCZ is only one of the factors affecting monsoon rainfall, and the dominance of each influential monsoon rainfall factor may be not the same in different areas within the monsoon region, which is why we have to compare multiple records. The synchronous variations between the relatively high smectite percentages and the northward shift of the ITCZ indicate that the northward movement of the ITCZ is the most important factor influencing the incremental changes in the river sediment load due to the increased smectite percentages in the Myanmar region. Here, we emphasize that the northward (southward) ITCZ shifts cause rainfall increases (decreases) relative to other rainfall drivers and that the changes in clay minerals reflect changes in rainfall and further sediment supply in the source area, which is a response to environmental changes. Regarding the influence of multiple factors on South Asian monsoon rainfall, more climate records and mutual verification of climate simulation work are needed, which is also one of the goals of our future research.

Technical corrections:

Q: Line 16: "millennial" instead of "millennium"
Re: Thank you; we have corrected these incorrect words.
Q: Line 18: Maybe "supply" or "transport" is better than "supplementation"
Re: Thank you; we have corrected these incorrect words.
Q: Line 20: exposure of Sunda Shelf?
Re: Thank you for your question. We did not express it clearly here. Here, we refer to the exposure of Andaman-Nicobar Island.
Q: Line 21 "millennial" instead of "millennium"
Re: Thanks; we corrected these inappropriate words.
Q: Line 23 "millennial" instead of "millennium"
Re: Thank you; we have corrected these incorrect words.
Q: Line 30: It would read better to replace "the paleoclimate and paleo-ocean" with "paleoclimate and paleoceanographic conditions"
Re: Thank you; we have corrected these incorrect words.
Q: Line 38: "important" might be a better word than "nonnegligible"
Re: Thank you; we have corrected these incorrect words.
Q: Line 53: I don't think statements like "discussed as a hot topic" add much to the context. Better to state which studies have discussed provenance in the BoB and the collective contribution of these studies.
Re: Thank you for your kind reminder. In the revision, we have reorganized the introduction section according to your comments and outlined the previous work done in the BoB area on the source of the terrigenous materials. We have modified it to the following statement: Previous studies have suggested that Himalayan material transported by the G-B River was the overwhelmingly dominant source of materials in the northern BoB during the Holocene (Li et al., 2018; Ye et al., 2020), and the main sources were the Indian Peninsula and Himalayan weathered material in the western BoB (Kessarkar et al., 2005; Tripathy et al., 2011; Tripathy et al., 2014). In the eastern BoB, the sediment sources are the Himalayan Range (transported by the G-B River), Indo-Burman Ranges and the Myanmar regions, through which the Irrawaddy River flows (Colin et al., 1999; Joussain et al., 2016). The terrigenous detrital material in the Andaman Sea are mainly Myanmar-origin sediments transported by the Irrawaddy River (Ali et al., 2015; Awasthi et al., 2014; Colin et al., 2006). A series of terrigenous problems, such as changes in the source area and the proportion of terrigenous matter in various regions of the BoB from

the Last Glacial Maximum to the Holocene, the distribution range of terrigenous materials in the Indian Peninsula on the west side of the BoB and in the Myanmar region on the east side, and how G-B River sediments migrated in the Bay of Bengal, are not yet clear.

Q: Line 65-66: Please explain more about why Ninetyeast Ridge is an ideal location- appropriate sed rates for a gravity core? And only receiving fine-grained hemipelagic sediments because it is bathymetrically above fan sedimentation?

Re: Thank you; we have revised it in the article. The core is located above the abyssal plain at ~900 m, and the channels on the eastern side of the BoB is at ~700 m. The channel near core 17I106 is dead (Curray et al., 2003) and does not support that large turbidity activity occurred there and provided vast amounts of clay minerals. The ~900 m height and approximately 400 km straight line distance between the core and active channel preliminarily indicate that turbidites are not the major provider compared to surface current input, and site 17I106 received fine-grained hemipelagic sediments from the surrounding area, which is an ideal area to understand paleoclimatic and paleoceanographic conditions in the BoB.

Q: Line 83: Centrifuged? What rpm and duration?

Re: Thank you for your reminder; we have added the complete experimental procedure. The sedimentation method involved placing the sample in a beaker with an inner diameter of 7 cm and a height of 10 cm at an experimental temperature of 19 ˚C. The sedimentation time was calculated as 4 hours and 10 minutes according to the Stokes formula, and the upper 5 cm of liquid was extracted, followed by centrifugation at 5000 rpm for 10 minutes. The smear was made into a natural slice, and the natural slice was heated in an oven at 60 ˚C for 24 hours to make a glycol-saturated slice for the subsequent test.

Q: Line 84: hydrochloric acid

Re: Thank you; we have corrected these incorrect words.

Q: Line 86: Were the slides treated with ethylene glycol? Were clay mineral standards used? Or just the Biscaye method?

Re: Thank you. These slides were treated with ethylene glycol and used only the Biscaye method.

Q: Line 92: Are these bulk sediments or a specific grain size fraction?

Re: Thank you for your question. It was an oversight not to note the sample information clearly; we use < 63μm sediments for Sr-Nd isotope experiments.

Q: Line 96: How were the Sr and Nd isolated? Info on columns, etc.

Re: Thank you for your suggestion. Our experimental method is not sufficiently described in the initial article, and we have revised it to "used the experimental method as described by Dou et al. (2016)". Carbonates were removed from 70 to 100 mg bulk powder samples by leaching with 0.25 N HCl for 24 h at 50 °C. The residues were then completely digested in high-pressure Teflon bombs using a $HCl + HNO_3 + HClO_4 + HF$ solution. Rb and Sr were separated in 2.5 N HCl using Bio-Rad AG50W-X12, 200–400 mesh cation exchange resin. Sm and Nd were separated in 0.15 N HCl using P507 cation exchange resin.

Q: Line 137: "cannot" is a strong word, but yes, kaolinite transport can be limited, but some regions of open ocean have substantial aeolian kaolinite

Re: Thank you; we have corrected these incorrect words.

Q: Line 148: I suggest "abyssal plain" instead of "the normal seafloor"

Re: Thank you; we have corrected these incorrect words.

Q: Line 149: turbidity currents

Re: Thank you; we have corrected these incorrect words.

Q: Line 169-170: Not just the narrowing of the straight but exposure of the continental shelves including the Sunda Shelf and all the way up to Myanmar. The relative exposure of 200 km from the current Irrawaddy delta

can affect how sediments are trapped on the shelf or delivered to the deep ocean.

Re: Thank you for your question. We have added a discussion to the revised manuscript. Large amounts of fluvial sediments are trapped in deltaic and shelf areas, especially coarse-grained sediments, but core 17I106 sediments were deposited after the long-distance transport of large amounts of fine-grained terrestrial material, and the fine-grained material in fluvial sediments can be transported over long distances. The 200 km change in shelf distance is not a large transport distance compared to the transport distance from the source area to core 17I106. Moreover, exposed shelf areas generally result in more weathered materials entering the deep sea, but the simultaneous decrease in smectite percentages from the Myanmar area as sea level decreases suggests that shelf denudation is not the main factor affecting our smectite record, and previous studies in the Andaman Sea also reveal no relevant effect on the alteration of terrestrial source material supply by exposed shelves (Ali et al., 2015; Awasthi et al., 2014).

Q: Line 182: "transport" is a better word choice than "importation"

Re: Thank you; we have corrected these incorrect words.

Q: Line 229: Please highlight any information about how far northward the ITCZ may have shifted before and after the LGM. Would this bring noticeable change to the Indo-Burma area?

Re: Thank you for your comments, and we have made further changes. Our smectite record shows that before the LGM, the ITCZ was in a relatively southerly position in the Myanmar area, while during the late LGM, the northward movement of the ITCZ in the Bay of Bengal led to increased rainfall in the Myanmar source area and an increased supply of smectite. At the same time, the ITCZ did not significantly shift in the Arabian Sea region in either the early LGM or late LGM, which is what the Arabian Sea ITCZ record shows (Deplazes et al., 2013).

Q: Line 245: Himalayas, mention the G-B transport.

Re: Thank you; we have corrected these incorrect words.

Q: Figure 5. Label the name of your core on the map.

Re: Thank you for the reminder. We have modified Figure 5 and added the core name to the map.

**References**

Abram, N.J., Hargreaves, J.A., Wright, N.M., Thirumalai, K., Ummenhofer, C.C., England, M.H., 2020. Palaeoclimate perspectives on the Indian Ocean Dipole. Quaternary Science Reviews 237.

Ali, S., Hathorne, E.C., Frank, M., Gebregiorgis, D., Stattegger, K., Stumpf, R., Kutterolf, S., Johnson, J.E., Giosan, L., 2015. South Asian monsoon history over the past 60 kyr recorded by radiogenic isotopes and clay mineral assemblages in the Andaman Sea. Geochemistry, Geophysics, Geosystems 16, 505-521.

Awasthi, N., Ray, J.S., Singh, A.K., Band, S.T., Rai, V.K., 2014. Provenance of the Late Quaternary sediments in the Andaman Sea: Implications for monsoon variability and ocean circulation. Geochemistry, Geophysics, Geosystems 15, 3890-3906.

Clemens, S.C., Yamamoto, M., Thirumalai, K., Giosan, L., Richey, J.N., Nilsson-Kerr, K., Rosenthal, Y., Anand, P., McGrath, S.M., 2021. Remote and local drivers of Pleistocene South Asian summer monsoon precipitation: A test forfuture predictions. Science Advances 7.

Colin, C., Turpin, L., Bertaux, J., Desprairies, A., Kissel, C., 1999. Erosional history of the Himalayan and Burman Ranges during the last two glacial-interglacial cycles. Earth and Planetary Science Letters 171, 647–660.

Colin, C., Turpin, L., Blamart, D., Frank, N., Kissel, C., Duchamp, S., 2006. Evolution of weathering patterns in the Indo-Burman Ranges over the last 280 kyr: Effects of sediment provenance on87Sr/86Sr ratios tracer. Geochemistry, Geophysics, Geosystems 7, n/a-n/a.

Curray, J.R., Emmel, F.J., Moore, D.G., 2003. The Bengal Fan: morphology, geometry, stratigraphy, history and processes. Marine and Petroleum Geology 19, 1191-1223.

Deplazes, G., Lückge, A., Peterson, L.C., Timmermann, A., Hamann, Y., Hughen, K.A., Röhl, U., Laj, C., Cane, M.A., Sigman, D.M., Haug, G.H., 2013. Links between tropical rainfall and North Atlantic climate during the last glacial period. Nature Geoscience 6, 213-217.

DiNezio, P.N., Puy, M., Thirumalai, K., Jin, F.-F., Tierney, J.E., 2020. Emergence of an equatorial mode of climate variability in the Indian Ocean. Science Advances 6.

DiNezio, P.N., Tierney, J.E., 2013. The effect of sea level on glacial Indo-Pacific climate. Nature Geoscience 6, 485-491.

Dou, Y., Yang, S., Shi, X., Clift, P.D., Liu, S., Liu, J., Li, C., Bi, L., Zhao, Y., 2016. Provenance weathering and erosion records in southern Okinawa Trough sediments since 28 ka: Geochemical and Sr–Nd–Pb isotopic evidences. Chemical Geology 425, 93-109.

Joussain, R., Colin, C., Liu, Z., Meynadier, L., Fournier, L., Fauquembergue, K., Zaragosi, S., Schmidt, F., Rojas, V., Bassinot, F., 2016. Climatic control of sediment transport from the Himalayas to the proximal NE Bengal Fan during the last glacial-interglacial cycle. Quaternary Science Reviews 148, 1-16.

Kessarkar, P.M., Rao, V.P., Ahmad, S.M., Patil, S.K., Anil Kumar, A., Anil Babu, G., Chakraborty, S., Soundar Rajan, R., 2005. Changing sedimentary environment during the Late Quaternary: Sedimentological and isotopic evidence from the distal Bengal Fan. Deep Sea Research Part I: Oceanographic Research Papers 52, 1591-1615.

Li, J., Liu, S., Shi, X., Zhang, H., Fang, X., Chen, M.-T., Cao, P., Sun, X., Ye, W., Wu, K., Khokiattiwong, S., Kornkanitnan, N., 2018. Clay minerals and Sr-Nd isotopic composition of the Bay of Bengal sediments: Implications for sediment provenance and climate control since 40 ka. Quaternary International 493, 50-58.

Pico, T., McGee, D., Russell, J., Mitrovica, J.X., 2020. Recent Constraints on MIS 3 Sea Level Support Role of Continental Shelf Exposure as a Control on Indo-Pacific Hydroclimate. Paleoceanography and Paleoclimatology 35.

Thirumalai, K., DiNezio, P.N., Tierney, J.E., Puy, M., Mohtadi, M., 2019. An El Niño Mode in the Glacial Indian Ocean? Paleoceanography and Paleoclimatology 34, 1316-1327.

Tierney, J.E., Pausata, F.S.R., deMenocal, P., 2015. Deglacial Indian monsoon failure and North Atlantic stadials linked by Indian Ocean surface cooling. Nature Geoscience 9, 46-50.

Tripathy, G.R., Singh, S.K., Bhushan, R., 2011. Sr-Nd isotope composition of the Bay of Bengal sediment impact of climate on erosion in the Himalaya Geochemical Joural 45, 175-186.

Tripathy, G.R., Singh, S.K., Ramaswamy, V., 2014. Major and trace element geochemistry of Bay of Bengal sediments: Implications to provenances and their controlling factors. Palaeogeography, Palaeoclimatology, Palaeoecology 397, 20-30.

Weldeab, S., Rühlemann, C., Ding, Q., Khon, V., Schneider, B., Gray, W.R., 2022. Impact of Indian Ocean surface temperature gradient reversals on the Indian Summer Monsoon. Earth and Planetary Science Letters 578.

Ye, W., Liu, S., Fan, D., Zhang, H., Cao, P., Pan, H.-J., Li, J., Li, X., Fang, X., Khokiattiwong, S., Kornkanitnan, N., Shi, X., 2020. Evolution of sediment provenances and transport processes in the central Bay of Bengal since the Last Glacial Maximum. Quaternary International.

---

## Author Comment (AC3)

**comments (black font) and responses (blue font)**

Reviewer #1

The topic of the paper appears suitable for CP. I consider both the overall scientific significance and quality to be good to fair and hence recommend minor to major revisions. A better understanding of latitudinal ITCZ migrations, both on orbital and millennial time scales is indeed desirable, and existing knowledge is either sparse or conflicting. The authors have worked extensively in the general area and topic before and provide sufficient evidence of a sound understanding on regional paleoceanographic change associated with shifts in the ITCZ.

To base their whole story around a single, short core (171106) and mainly one property (smectite) with the additional, occasional provenance data (Sr-Nd) is a little thin in my eyes. The chronology is sound and includes age modeling and uncertainty evaluation. The same appears to be the case for the clay mineralogy and provenance tracers.

Re:Thank you for your comment. The percentages of illite and smectite added up to 78.5% and these two clay minerals have a reverse-phase variation, which separately represents the sediment input of the G-B and Irrawaddy Rivers. Thus, smectite can represent the main characteristics of clay minerals. Sr-Nd isotopes are generally used for tracing sources of terrestrial sediment, especially during long-term analysis. Although core 17I106 is relatively short and the average resolution is approximately 256 yr/cm, it can be utilized to indicate paleoclimate and paleoceanographic condition changes on the millennial scale. In summary, clay minerals and Sr-Nd isotopes of the studied core are sufficient to distinguish sediment sources in the study area, and the clay mineral record is suitable to trace the ITCZ shift on the millennial scale, which has little influence on our main conclusions.

The whole discussion on smectite suffers, in my opinion, from the fact that the record is too short (MIS 3-1) and does not cover a full glacial-to-interglacial cycle. One cannot see how the response was, for instance, during MIS 6 and 5, which would be critical to know here. Any discussion of orbital variability is hence hampered. This is specifically true for any statement implicating changes in sea-level and and their effect on changing provenance and clay mineralogy (availability or lack of accommodation space on the shelves). The core is apparently only 162 cm long - why? Wasn't there a longer alternative to conduct such a study?

The same principal problem surrounds the discussion of the smectite peak around the LGM (21-18 ka). If there were a record for termination 2 (MIS 6/5 transition) one could see if there are common rules established during glacial maxima that are either regional or not. MIS 6 and termination 2 have comparable orbital configurations relative to the LGM and termination 1, hence possible shifts of the ITCZ – which are invoked in the discussion – should have been quite similar, at least from a global view, from which regional deviations could then be derived or discussed if present.

Re:Thank you for your comment. This advice is useful inspiration for us to plan future work; our group has collected core sediments annually in the Bay of Bangel since the first trip in 2010. However, thus far, on the Ninetyeast Ridge, we collect only two cores that were both shorter than 200 cm; in the next several years, we will focus on this area. The discussion of orbital variability is certainly critical

for the mechanisms of South Asian monsoon evolution and sea level change,  but unfortunately, it is impossible to continue the related work on this core due to the lack of longer gravity core(s). Certainly, MIS 6 and termination 2 have comparable orbital configurations relative to those of the LGM and termination 1, and similar orbital configurations may induce similar regional shifts in the ITCZ according to the view that orbital forces drive the thermal gradients. We also hope to perform more work on long time scales in the future and discuss shifts in the ITCZ on orbital time scales.

There is the complete lack of mentioning turbidity currents, which provide the vast majority of sediments from the river mouth of the Swatch of No Ground (SoNG) to the BoB and neighboring regions. Even though the short core presented here seems to show a rather steady deposition, plumes from turbidity currents will reach the site and affect the clay mineralogy. In this context, it is also odd that none of the IDOP-Expedition 354 studies, which are in close proximity to the core cite, have been cited ore discussed.

This is important and needs clarification and elaboration. How far above the surrounding is the core site? As mentioned above the authors completely ignore the possibility of turbidite deposition. Even if only the fine-grained upper sediment clouds of distal turbidites (those naturally contain a high percentage of suspended fine material, i.e., clay minerals) reach the sediment site, it would have a large effect on the clay mineralogy of site 171106. After all, the sites in not far from the channels on the eastern side of the BoB.

Re: Thank you for your comment. Turbidity currents can provide vast amounts of sediments from the river mouth of the Swatch of No Ground (SoNG) to the BoB and neighboring regions, especially near active channels, which is confirmed by geomorphological observations (Curry et al., 2003) and clay mineral studies of surface sediments (Sun et al., 2020). We modified Figure 1 and added two profiles to indicate the location of the core in relation to the channels. As the following figure shows, the study core is located above the central BoB seafloor at ~900 m and the channels on the eastern side (E6) of the BoB at ~700 m. The channel near core 17I106, named E6, is dead, has lacked massive landslides during the past 45 ka (Curry et al., 2003) and does not support that large turbidity activities have occurred there and provided a vast amount of clay minerals. Moreover, the ~900 m height of the seamount and the nearest 400 km straight line distance between the core and SoNG hindered the entry of fine-grained upper sediment clouds of distal turbidites into the core site and caused the turbidites be a minor provider compared to surface current input.  IODP Expedition 354 studies provide very important background research. We mistakenly forgot to cite related references in the writing process, and we will add them in the revision.

[Figure]

New Figure. 1 Geographical setting of the BoB. The locations of cores 17I106 (red asterisks) and U1455 (orange diamond) are shown. The Bengal Fan diagram is modified from Curray et al. (2003). The white dashed lines outline the scales of the Bengal Fan and the Nicobar Fan. The solid gray lines and black letters represent the turbidity channel and the reference names of the principal channels, respectively. The pink solid line is the 'active' channel. The heavy dashed line represents the shelf edge and bathymetric highs. The dotted-dashed line is the outline of the most recently active subfan. The solid white lines represent the two profile positions, which are shown on the right along with the depth legend of Figure 1.

I also miss the discussion on the variability of the Oxygen Minima Zona (OMZ) when the authors invoke the connection to the Northern Hemisphere. What would be the consequences for the area of the core site? For the Arabian Sea, Schulz et al., (1998, Nature) clearly linked the millennial-scale coupling to Greenland to shifts in extent of the OMZ, pointing to changing water mass composition and oxygenation.

Also, the authors mark H1-H4 and discuss the relation to smectite variability. The data does not show that in my opinion. Even if slight shifts are employed to account for potential mismatches in the age model, there is no consistent relationship, i.e. the various clay minerals occur either at the highs, lows or transitions of H events.

Re:Thank you for your comment. We note that Schulz et al. (1998, Nature) suggested a lowered southwestern monsoonal intensity following low total organic carbon (TOC) percentages, which showed that weak summer monsoonal productivity is associated with intervals of high-latitude atmospheric cooling and injection of melt water into the North Atlantic basin. The oxygenation of bottom waters is directly influenced by summer monsoonal productivity, resulting in strong variations in the intensity of the OMZ during the last glacial period. However, in the Bay of Bengal, Zhou et al. (2021) suggested that during Heinrich Stadial 1 and the Younger Dryas, i.e., when the AMOC collapsed, weaker South Asian precipitation diminished stratification and enhanced primary productivity. Thus, the response of marine productivity to monsoons, which receive remote forces

from the North Atlantic, was likely different between the BoB and Arabian Sea. Based on our understanding, the variability in the OMZ is mainly connected with the intensity of the summer monsoon. However, the consequence of OMZ variety on the Ninetyeast Ridge is an interesting idea that inspires us to take the next step in our work.

For the OMZ in the BoB and Arabian Sea, McCreary et al. (2013) suggested that the OMZ in the BoB is weaker than that in the Arabian Sea because this bay lacks a remote source of detritus from the western boundary. Although detritus has a prominent annual cycle, the model OMZs do not show a similar cycle because there is not enough time for significant remineralization to occur. Then, the OMZ in the BoB may not show the same variations as the OMZ in the Arabian Sea because lower biological oxygen consumption is also assumed to be responsible for a less intense OMZ in the BoB (Rixen et al., 2020).

Although clay minerals play important roles in organic carbon transport, as suggested by Blattmann et al. (2019), the connection between marine organisms that may influence the OMZ and clay minerals does not significantly impact on the transport and deposition of clay minerals. To explain the relationship between our clay mineral record and the Atlantic force, we note the inappropriateness of the intended meaning of out expression. We focus on the source area that is responsible for physical and chemical weathering due to decreased rainfall and temperatures during the North Atlantic cold-climate periods (Heinrich events and YD period, Figure 3h), specifically, the increasing (decreasing) trend of illites (smectites) just before Heinrich events and the decreasing (increasing) trend of illites (smectites) after Heinrich events.

Another lack of discussion surrounds the length of the core, which is only 162 cm. The sample resolution of 1 cm implies a 300-yr resolution, however, bioturbation should mix the sediment over several cms and smear according ages. What are the author's assessment of this effect and how would it affect their conclusions?

Re:Thank you for your comment. Indeed, bioturbation is a common phenomenon during the deposition of seafloor sediments, and we cannot exclude the disturbance of bioturbation absolutely. However, we think it is more likely that the shift in the ITCZ does influence the clay mineral addition than the bioturbated mixed clay minerals, and it is difficult for bioturbation from different periods to cause such a consistent and coordinated smectite percentage change. There is no significant mixing within the chronostratigraphic framework, and benthic foraminifera are less abundant compared to other cores in the BoB (unpublished); thus, the disturbance caused by benthic organisms may be relatively small.

More specific comments:

Fig. 3
Sr/Nd resolution is too low to determine temporal variations. The only real change happened at 14-15 ka, probably as a result to changes surrounding Meltwater Pulse 1A, which are indicative of a major re-organization of the global thermohaline circulation. However, the data shows that the glacial-to-interglacial and millennial-scale sources likely did not change. Why is that?

Re:Thank you for your comment. Certainly, the Sr/Nd resolution in our core is too low to determine temporal variations, especially before the LGM, and Sr/Nd isotopes do not fully represent glacial-interglacial and millennium-scale changes. In our record, the difference between clay minerals and Sr/Nd isotopes may be consistent with the viewpoint that clay minerals may be transported over long distances, while coarser terrestrial sediments indicated by Sr/Nd isotopes can be transported only to more proximate locations. We suggest that these nearby coarse sources are not significantly affected by climate and environmental driving forces, while clay minerals mainly from the larger G-B River and Myanmar source areas affected by climate driving forces are more likely to show glacial-interglacial and millennial-scale changes.

In this context, Fig. 1 is missing a depth legend for both the marine and terrestrial elevations. Figures 1 and 5 could potentially be combined into a single figure.

Re:Thank you for your careful review. We combined Figures 1 and 5 into the new Figure 1 above and added an elevation legend to it.

**References**

Blattmann T M , Liu Z , Zhang Y , et al. Mineralogical control on the fate of continentally derived organic matter in the ocean[J]. Science, 2019, 366(6466): eaax5345.

Schulz,, H., von Rad,, U., Erlenkeuser,, H. et al. Correlation between Arabian Sea and Greenland climate oscillations of the past 110,000 years. Nature 393, 54–57 (1998). https://doi.org/10.1038/31750

Curray, J.R., Emmel, F.J., Moore, D.G., 2003. The Bengal Fan: morphology, geometry, stratigraphy, history and processes. Marine and Petroleum Geology 19, 1191-1223.

McCreary, J.P., Yu, Z., Hood, R.R., Vinaychandran, P.N., Furue, R., Ishida, A., Richards, K.J., 2013. Dynamics of the Indian-Ocean oxygen minimum zones. Progress in Oceanography 112-113, 15-37.

Rixen, T., Cowie, G., Gaye, B., Goes, J., do Rosário Gomes, H., Hood, R.R., Lachkar, Z., Schmidt, H., Segschneider, J., Singh, A., 2020. Reviews and syntheses: Present, past, and future of the oxygen minimum zone in the northern Indian Ocean. Biogeosciences 17, 6051-6080.

Sun, X., Liu, S., Fang, X., Li, J., Cao, P., Zhao, G., Khokiattiwong, S., Kornkanitnan, N., Shi, X., 2020. Clay minerals of surface sediments from the lower Bengal Fan: Implications for provenance identification and transport processes. Geological Journal 55, 6038-6048.

Zhou, X., Duchamp-Alphonse, S., Kageyama, M., Bassinot, F., Beaufort, L., Colin, C., 2020. Dynamics of primary productivity in the northeastern Bay of Bengal over the last 26 000 years. Climate of the Past 16, 1969-1986.